

# Comprehensive genomic characterisation of the NAC transcription factor family and its response to drought stress in *Eucommia ulmoides*

Qi Wang[1], FengCheng Hu[2], ZhaoQun Yao[3], XinFeng Zhao[2], GuangMing Chu[1] and Jing Ye[1]

[1] Laboratory of Forestry Department, Agricultural College, Shihezi University, Shihezi, China
[2] Lveyang County Forest Tree Seedling Workstation, Forestry Bureau of Lveyang County, Lveyang, China
[3] Laboratory of Plant Protection Department, Agricultural College, Shihezi University, Shihezi, China

## ABSTRACT

The NAC transcription factor family enhances plant adaptation to environmental challenges by participating in signalling pathways triggered by abiotic stressors and hormonal cues. We identified 69 *NAC* genes in the *Eucommia ulmoides* genome and renamed them according to their chromosomal distribution. These EuNAC proteins were clustered into 13 sub-families and distributed on 16 chromosomes and 2 scaffolds. The gene structures suggested that the number of exons varied from two to eight among these *EuNACs*, with a multitude of them containing three exons. Duplicated events resulted in a large gene family; 12 and four pairs of *EuNACs* were the result of segmental and tandem duplicates, respectively. The drought-stress response pattern of 12 putative *EuNACs* was observed under drought treatment, revealing that these *EuNACs* could play crucial roles in mitigating the effects of drought stress responses and serve as promising candidate genes for genetic engineering aimed at enhancing the drought stress tolerance of *E. ulmoides*. This study provides insight into the evolution, diversity, and characterisation of *NAC* genes in *E. ulmoides* and will be helpful for future characterisation of putative *EuNACs* associated with water deficit.

## INTRODUCTION

*Eucommia ulmoides* Oliver is a highly valued tertiary relict perennial dicotyledonous tree species native to China (*Deng et al., 2022*). It is widely used in industry because it not only produces wood but is also a valuable raw biomaterial for extracting the active ingredients of Chinese medicine and *trans*-rubber (*Wei et al., 2021*; *Zhu & Sun, 2018*). Compared with *cis*-rubber, *trans*-rubber has unique characteristics, such as high hardness, resistance to acid and alkali corrosion, good insulation, and a low thermal expansion/contraction coefficient (*Du et al., 2023*; *Enoki, Doi & Iwata, 2003*; *Kent & Swinney, 1966*; *Rose & Steinbuchel, 2005*). To improve the viscosity, resilience, elasticity, weather resistance, and tensile strength of *trans*-rubber, it can be made into *cis*-rubber by vulcanisation (*Yan, 1995*). The

Corresponding authors
GuangMing Chu, chgmxj@163.com
Jing Ye, yejing_shz@foxmail.com

vulnerability of *Hevea brasiliensis* to pests and diseases, as well as its narrow habitat, has led to significant challenges for the rubber industry (*Tang et al., 2016*). *E. ulmoides* has wide adaptability, few pests and diseases, and its leaves, bark, and pericarp are rich in *trans*-rubber. Therefore, *E. ulmoides* is considered an ideal alternative or complementary tree species to *H. brasiliensis* (*Guo et al., 2023*; *Wuyun et al., 2018*). During the growth of *E. ulmoides*, some environmental factors, such as drought and low-temperature stress, can prevent its full genetic potential, resulting in reduced yield and even plant death (*Zuo et al., 2022*). The identification and utilisation of resistance genes is the basis for breeding resistant varieties. Transcription factors (TFs), which activate or inhibit their expression by specifically binding to *cis*-acting elements on the promoters of target genes, play an important role in many biological processes (*Oksuz et al., 2023*; *Yuan et al., 2020*). As a plant-specific supergene family, *NAC* has been demonstrated to play a key role in plant growth and development and in the response to abiotic stress (*Du et al., 2022*; *Hussain et al., 2017*). Notably, *NAC* is very important in plant adaptation to land (*Xu et al., 2014*). Therefore, *NAC* family genes have been widely studied in many species. However, the identification and analysis of *NAC* genes in *E. ulmoides* have not been emphasised.

The *NAC* (no apical meristem (NAM), *Arabidopsis* transcription activator (ATAF1/ATAF2), and cup-shaped cotyledon (CUC2)) family is one of the largest gene families in plants. The N-terminus region has a highly conserved NAM domain, and the C-terminus consists of variable transcriptional regulatory regions, the latter of which have been implicated in specific biological functions (*Shao, Wang & Tang, 2015*). Increasing evidence suggests that NAC TFs have multiple functions in plant-responses to biotic and abiotic stress. *SlNAC1* is involved in the process of fruit softening and fruit pigmentation based on the phytohormone pathway (*Ma et al., 2014*). *NAC13* has important significance in popular responses to salt stress (*Zhang et al., 2019*). In wheat, overexpression of *TaNACL-D1* and *TaNAC071-A* improves resistance to *Fusarium* head blight (*Perochon et al., 2019*) and drought (*Mao et al., 2022*), respectively; *TaNAC30* negatively regulates stripe rust (*Wang et al., 2018*). Furthermore, *TaNAC29* improves salt tolerance by strengthening the antioxidant system (*Xu et al., 2015*). *GhirNAC2* regulates ABA biosynthesis and stomatal closure by regulating *GhNCED3a/3c* expression, thus playing an active role in cotton drought resistance (*Shang et al., 2020*). Although NAC TFs are related to various developmental processes and stress responses in plants, the specific functions of most *NAC* genes remain obscure, especially in *E. ulmoides*.

The chromosome-level genome of *E. ulmoides* was recently sequenced (*Li et al., 2020*); this provides the opportunity to systematically study the *NAC* gene family and to explore the potential functional involved in *E. ulmoides* biotic and abiotic responses. In the present study, we performed genome-wide identification and characterisation of NAC proteins based on the genome of *E. ulmoides*. In addition, we surveyed their expression under drought stress by transcriptome sequencing. This study will lay the foundation for further studies of the molecular mechanisms of NAC TFs in *E. ulmoides* response to drought stress.

## MATERIALS & METHODS

### Identification of EuNAC proteins from the *E. ulmoides* genome

The complete genome data of *E. ulmoides* were obtained from the Gene Warehouse (https://ngdc.cncb.ac.cn/gwh/Assembly/25206/show). The NAC protein sequences of *Arabidopsis thaliana* were obtained from *Arabidopsis* Information Resources (TAIR, https://www.arabidopsis.org/index.jsp), and the protein sequences of *poplar* and *Oryza sativa* were both derived from the Ensembl Plants website (http://plants.ensembl.org/index.html). The Hidden Markov model (HMM) files of the NAC domain (PF01849) and the NAM domain (PF02365) were obtained from the Pfam database (https://pfam.sanger.ac.uk), which were used for identification analysis. HMMER 3.3.2 (http://hmmer.org/) was then employed to scan the NAC proteins from the *E. ulmoides* genome with the default parameters. The candidate *EuNACs* were further validated by the NCBI Conserved Domain Search Service (CD Search) (https://www.ncbi.nlm.nih.gov), SMART (http://smart.embl-heidelberg.de), and Pfam database. Proteins without NAC and NAM domains and duplicates were manually deleted. The molecular weight (MW) and isoelectric point (pI) of each protein were analysed using the ExPASy pI/Mw tool (https://www.expasy.org).

### Phylogenetic analysis of EuNAC proteins

The *A. thaliana* NAC protein sequences were downloaded from the TAIR database (http://www.Arabidopsis.org). Full-length protein sequence multiple alignments were performed using the ClustalW programme (*Larkin et al., 2007*). MEGA 6.0 software (*Hall, 2013*) was employed to construct an unrooted phylogenetic tree of *E. ulmoides* and *A. thaliana* NAC proteins using the neighbour-joining (NJ) method with 1,000 bootstrap iterations. All EuNAC proteins were classified according to the NAC protein classification criteria in *A. thaliana* (*Ooka et al., 2003*).

### Conserved motif and gene structure analysis of *EuNAC* genes

The Gene Structure Display Server (GSDS; http://gsds.cbi.pku.edu.cn/) programme was used to explore the exon/intron structure pattern of the *EuNAC* genes by comparing their predicted coding sequence with the corresponding full-length gDNA sequence. Multiple Expectation Maximization for Motif Elicitation (MEME) (http://meme-suite.org/) programmes were employed to identify the conserved domains for candidate EuNAC proteins with default parameters. The conserved motifs and exon/intron structure were visualised using Tbtools (*Chen et al., 2020*).

### Genome distribution, selective pressure, and synteny analysis of *EuNAC* genes

The location of each *EuNAC* gene on the chromosome was determined based on the *E. ulmoides* genome annotation file and visualised using TBtools software (*Chen et al., 2020*). MCScan X software (*Wang et al., 2012*) with default parameters was employed to analyse duplication events, and the intra-species and inter-species collinearity relationships. Circos software (*Krzywinski et al., 2009*) and TBtools were used for visualisation. TBtools were also used to calculate the nonsynonymous (*Ka*) and synonymous (*Ks*) rates of *EuNAC*

homologous genes. The selection pressure acting on the gene pairs was calculated based on the $Ka/Ks$ ratio, and the dates of each duplication event were further deduced with the formula T = Ks/2 $\lambda$, the mean synonymous substitution rate ($\lambda$) was assumed to be $6.5 \times 10^{-9}$ (*Liu et al., 2021*; *Lynch & Conery, 2000*).

### Promoter region analysis of *EuNAC* genes

The 2-kb promoter sequences upstream of the *EuNAC* genes start codon (ATG) were extracted, and the cis-acting elements and their potential related functions were predicted with the PlantCARE online server (http://bioinformatics.psb.ugent.be/webtools/plantcare/html/). The 20 cis-acting elements with the highest frequency were visualised using Tbtools (*Chen et al., 2020*).

### Expression analysis of *EuNAC* genes under drought stress

Two-year-old 'Qinzhong 1' grafted potted plants with consistent growth were placed in the Agricultural College of Shihezi University and well managed in the natural environment. Three months later, they were subjected to drought stress treatment. For drought treatment, the soil was saturated with water, and watering was terminated. Leaves were collected at 0, 15, 30, and 45 d and labelled CK, D15, D30, and D45, respectively. Each sample was pooled from six individual plants, and three biological replicates were set for each treatment. Samples were quickly cleaned with distilled water, immediately frozen in liquid nitrogen, and stored at −80 °C for future use.

The above samples were submitted to Beijing Novogen Bioinformatics Technology Co., Ltd. (Beijing, China) for cDNA library construction and transcriptome sequencing. The data were uploaded to the National Center for Biotechnology Information (NCBI) Sequence Read Archive, with accession number PRJNA958614. Gene expression levels were normalised with FPKM (fragment per kilobase per million mapped reads) and visualised with TBtools.

## RESULTS

### Identification of EuNAC proteins from the *E. ulmoides* genome

We identified 69 *NAC* genes in the *E. ulmoides* genome, named *EuNAC1* to *EuNAC69*, according to their order on the chromosomes (Table S1). The EuNAC proteins varied significantly in length and molecular weight. The length of proteins encoded by *EuNAC* genes ranged from 86 (*EuNAC50*) to 617 (*EuNAC64*) amino acids (*Jeong et al., 2008*) and the molecular weight varied from 9.81 (*EuNAC50*) to 70.46 kDa (*EuNAC64*); the isoelectric points (pIs) ranged from 4.51 (*EuNAC59*) to 10.01 (*EuNAC2*). This study also analysed other basic information about 69 *NAC* genes, including homologous genes in *A. thaliana*, open reading frame (ORF) length, location coordinates, chromosomal positions, and exon numbers (Table S1).

### Phylogenetic analysis and classification of EuNAC proteins

To investigate the evolutionary relationships among *EuNAC* family genes, MEGA6.0 software was employed to construct a neighbour-joining phylogenetic tree based on

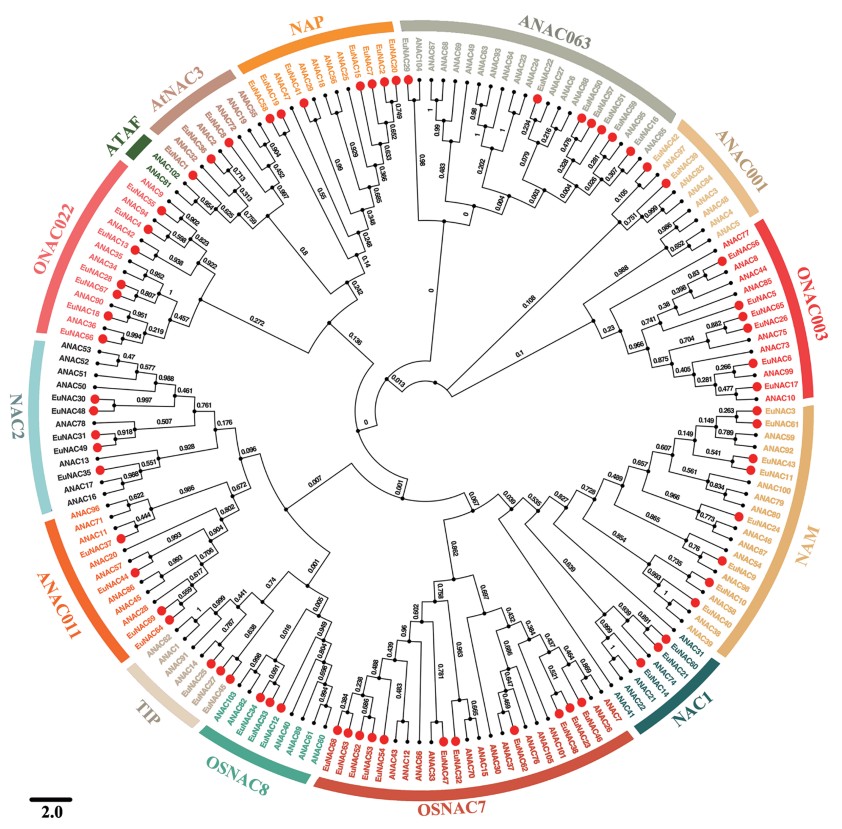

**Figure 1** **Phylogenetic relationships among *NACs* identified in *E. ulmoides* and *Arabidopsis thaliana*.** The unrooted phylogenetic tree was constructed by MEGA 6.0 software using the Neighbor-Joining (NJ) method with 1,000 bootstrap iterations. Each subfamily was distinguished by different colors.

full-length protein sequences of *NAC* in *E. ulmoides* and *A. thaliana* (Fig. 1) (*Hall, 2013*). According to the *NAC* classification system for *A. thaliana*, *NACs* from *E. ulmoides* were divided into 13 distinct subfamilies, namely *NAC2*, *ONAC022*, *AtNAC3*, *NAP*, *ANAC063*, *ANAC011*, *ONAC003*, *NAM*, *NAC1*, *OSNAC7*, *OSNAC8*, *TIP*, and *ANAC011*, using a phylogenetic tree; however, no *EuNAC* members were identified in the *ATAF* subfamily. Of these 13 subfamilies, *OSNAC7* had 11 *EuNAC* members, which was the most abundant, followed by *NAM*, which had eight members. *ANAC001* was the least frequent with only two members. Phylogenetic analysis revealed that EuNAC proteins had evolved in some diversity, similar to a report in *A. thaliana* (*Ooka et al., 2003*).

## Conserved motif and gene structure analysis of *EuNAC* genes

To gain insight into the functional regions of EuNACs, the conserved motifs for each EuNAC protein were analysed using the MEME programme. A total of 10 conserved motifs were identified and named motifs 1–10 (Table S2). These conserved motifs had large variations in length, with a distribution range of 11–50 amino acid residues. As shown in Fig. 2A, motif 3 had the highest frequency in the *EuNAC* family, and it existed in almost all members except *EuNAC2*, *19*, and *33*. In addition, motifs 1, 2, 4, 5, and 6

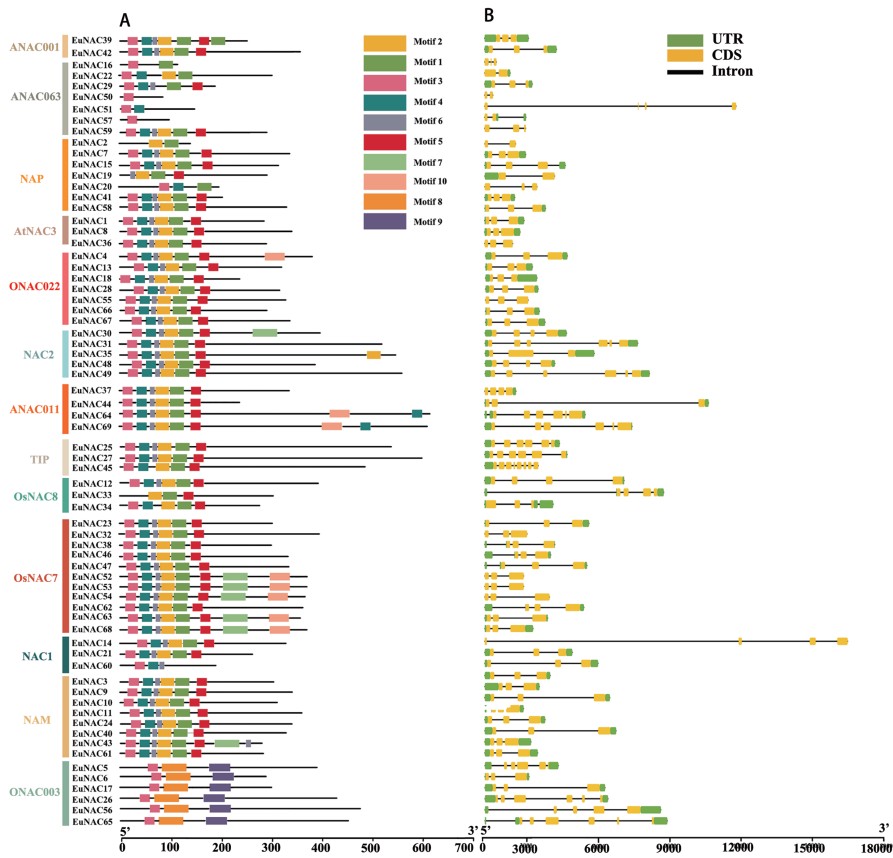

**Figure 2** **Motif compositions and DNA structures of *NAC* gene family in *E. ulmoides*.** (A) The conserved motif distribution of EuNAC proteins. Different motifs were distinguished by different colored boxes, and black lines represent non-conserved regions. (B) Gene structure of the *EuNAC* gene. Green boxes represent non-coding regions, yellow boxes represent exons, and black lines represent introns.

were very abundant in the *EuNAC* family, but none of the *ONAC003* subfamily members had these motifs. Most of the conserved motifs were distributed in the N-terminus of the NAC proteins, indicating that the N-terminal region plays an important role in *NAC* gene function. In addition, similar motif compositions existed among different members of the same subfamily, indicating that members of the same subfamily had similar functions.

To investigate the structural features of *EuNACs*, we analysed the intron/exon distribution patterns of each *EuNAC* gene. The exon distribution within the *EuNAC* genes varied from two to eight (Table S1, Fig. 2B). Forty-five (65.2%) genes had three exons, 10 (14.5%) genes had six exons, and *EuNAC45* had the largest number of exons, with eight exons (Fig. 2B, Table S1). Forty-nine (71.0%) *EuNAC* genes possessed less than three exons, indicating a low structural diversity among *EuNAC* genes.
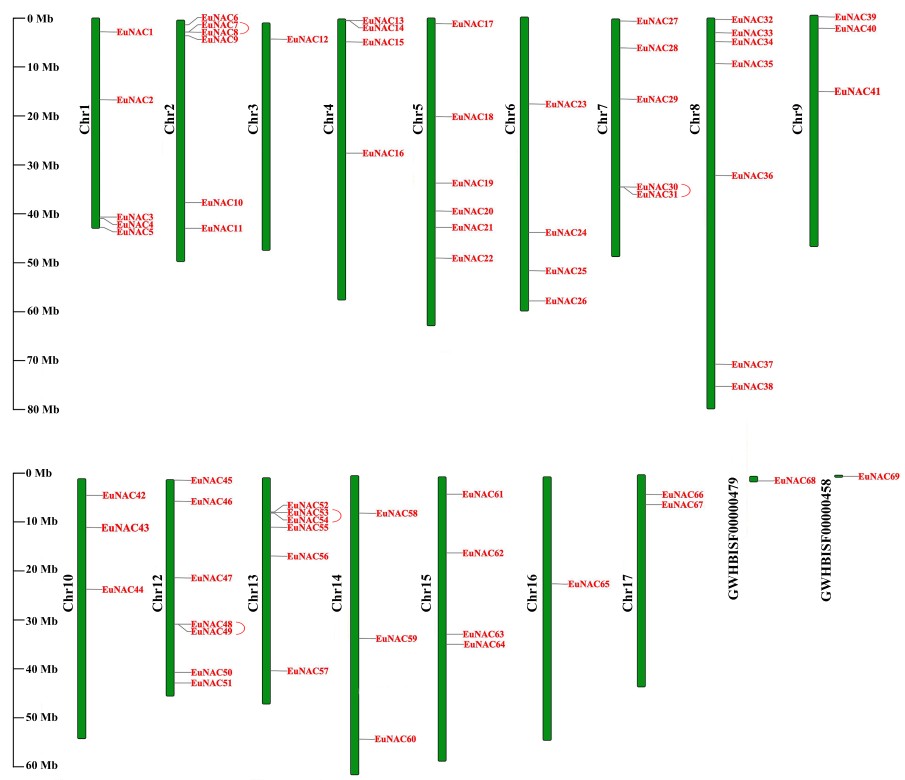

**Figure 3** **Distribution of 69 *EuNAC*s on 16 chromosomes and two scaffolds.** Vertical bars represent the chromosomes of *E. ulmoides*. The chromosome number is on the left of each chromosome. The scale on the left represents the length of the chromosome.

## Genome distribution, selective pressure, and synteny analysis of *EuNAC* genes

To investigate the distribution of *EuNAC* genes on the chromosomes of *E. ulmoides*, TBtools software was employed to map the chromosome locations for all *EuNACs* identified in this study (Fig. 3). The 69 *EuNACs* were unevenly scattered on 16 chromosomes and two scaffolds, and the length of each chromosome showed no correlation with the number of genes contained. Chromosomes 8 and 12 had the most *EuNACs*, both with seven genes. Only one *EuNAC* was distributed on chromosomes 3 and 16, and no *EuNACs* were distributed on chromosome 11. Notably, most *EuNACs* were distributed near the ends of the chromosome.

Furthermore, the duplication events of *EuNAC* gene family members were examined using MCScanX software. A total of 12 pairs of segmental duplications were identified in the *EuNAC* family, which were distributed across 15 chromosomes, except for chromosomes 3 and 11, and four pairs of tandem replications (*EuNAC7/8*, *EuNAC30/31*, *EuNAC48/49*, and *EuNAC53/54*) were identified, which were distributed on chromosomes 2, 7, 12, and 13, respectively (Fig. 4, Table S3). The results showed that segmental duplication events might be the crucial driving force in *EuNAC* gene family expansion. To evaluate the selection pressure of *EuNACs*, the *Ka*, *Ks*, and *Ka/Ks* for duplicated gene pairs were calculated. In

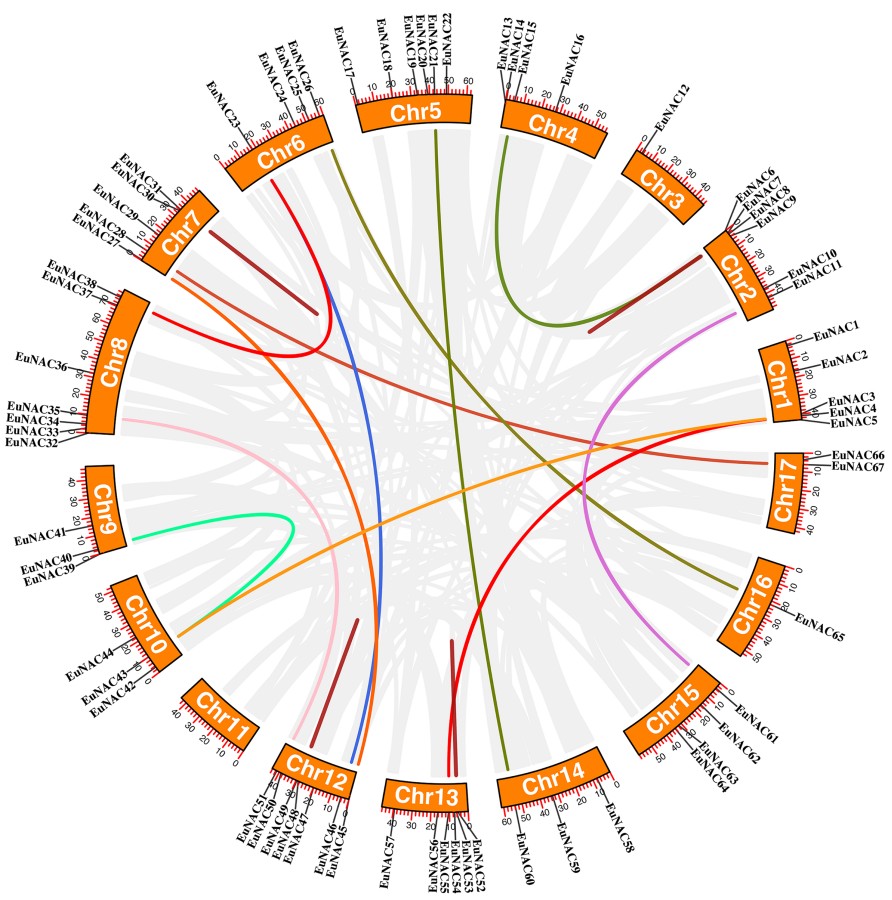

**Figure 4  Schematic representations of the interchromosomal relationships of *EuNAC* genes.** The deep red line represents the *EuNAC* gene pairs replicated in tandem, while the remaining colored lines represent the *EuNAC* gene pairs replicated in segments.

general, a *Ka/Ks* > 1 indicates positive selection, *Ka/Ks* = 1 indicates neutral selection, while *Ka/Ks* < 1 indicates purifying selection (*Vahdati & Lotfi, 2013*). The Ka/Ks ratios of 16 replicated *EuNAC* gene pairs were all less than 1, indicating that the evolution of the *EuNAC* gene family was subjected to purification selection (Table S4).

To further understand the phylogenetic mechanisms of the *EuNAC* gene family, we constructed a comparative homologous map of *NAC* genes in *E. ulmoides*, *A. thaliana*, and *Oryza sativa*. In total, 31 *EuNACs* had a collinear relationship with 27 *AtNACs* and 10 *OsNACs*. Thirty-one and eleven pairs of *NAC* homologous gene pairs were formed between *E. ulmoides* and *A. thaliana* and between *E. ulmoides* and *Oryza sativa*, respectively (Fig. 5, Table S5). The results indicate that the *NAC* genes underwent significant evolution and replication after differentiation in monocotyledonous and dicotyledonous plants.

## Promoter region analysis of *EuNAC* genes

To investigate the potential functions of *EuNACs*, we employed PlantCARE to predict the cis-acting elements within the 2.0 kb sequence upstream of the initiation codon (ATG) of

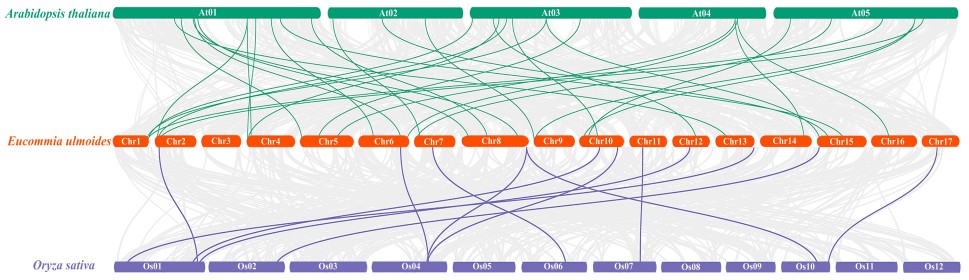

**Figure 5** Synteny analysis of *NAC* genes between *E. ulmoides* and two representative plant species (***Arabidopsis thaliana*** **and** ***Oryza sativa***). Green and purple lines represent syntenic *NAC* gene pairs of *E. ulmoides* and *A. thaliana* and *O. sativa*, respectively.

*EuNACs* (Table S6). As expected, both TATA and CAAT boxes with good characteristics were found in the results; we also found several other CIS regulators (Table S7 and Fig. S1). They were mined in the promoter region of *EuNAC*s. As shown in Fig. 6, we divided the homeopathic elements into four categories according to their functions. The first category was phytohormone-responsive elements, such as CGTCA-motif, TGACG, TCA-element, and ABRE, wherein ABREs have been associated with ABA responses and TCA-element have been associated with salicylic acid responsiveness. The second category was elements of cis-regulation related to the response to external or environmental pressure. This category includes low-temperature response elements (LTR), which respond to external abiotic stress, abundant cis-regulatory elements (*Jeffares, Penkett & Bähler, 2008*), which are required for anaerobic induction, and MYB binding site (MBS) elements. Notably, 29 of the 69 *EuNAC* promoters contained MBS elements that were involved in drought induction as MYB binding sites and could be predicted based on their responses to drought stress treatments. The third category included light-responsive elements, such as G-box, Box-4, and GT1-motif, in which at least one photo-responsive element was detected in almost every promoter region of *EuNACs*. The last category was cis-regulatory elements related to growth and development. CAT-box and O2-site were mainly detected, which also indicated that most *EuNACs* may be involved in *E. ulmodies* meristem expression, zein metabolism regulation, and cell cycle regulation. Finally, based on the above results, *EuNACs* may be involved in stress response, light, hormones, and growth pathways.

## Expression analysis of *EuNAC* genes under drought stress

To further investigate the potential function of *EuNACs* in response to drought stress, comparative transcriptomics of *E. ulmoides* under drought stress were used to analyse the expression patterns of *EuNACs*. Of the 69 *EuNAC* genes, 20 showed high expression levels with FPKM ≥ 20, including *EuNAC1*, *2*, *12*, *15*, *20*, and *25*. Forty-three *EuNACs* showed low expression levels, with FPKM ≤ 20, including *EuNAC10*, *11*, *13*, *14*, *16*, *17*, and *18*. In addition, *EuNAC19*, *47*, *52*, *54*, *55*, and *58* were not expressed in *E. ulmoides* leaves (Table S8). Differential gene expression (DEG) analysis showed that 12 *EuNAC* genes were significantly differentially expressed, of which *EuNAC2*, *3*, *11*, and *14* were upregulated

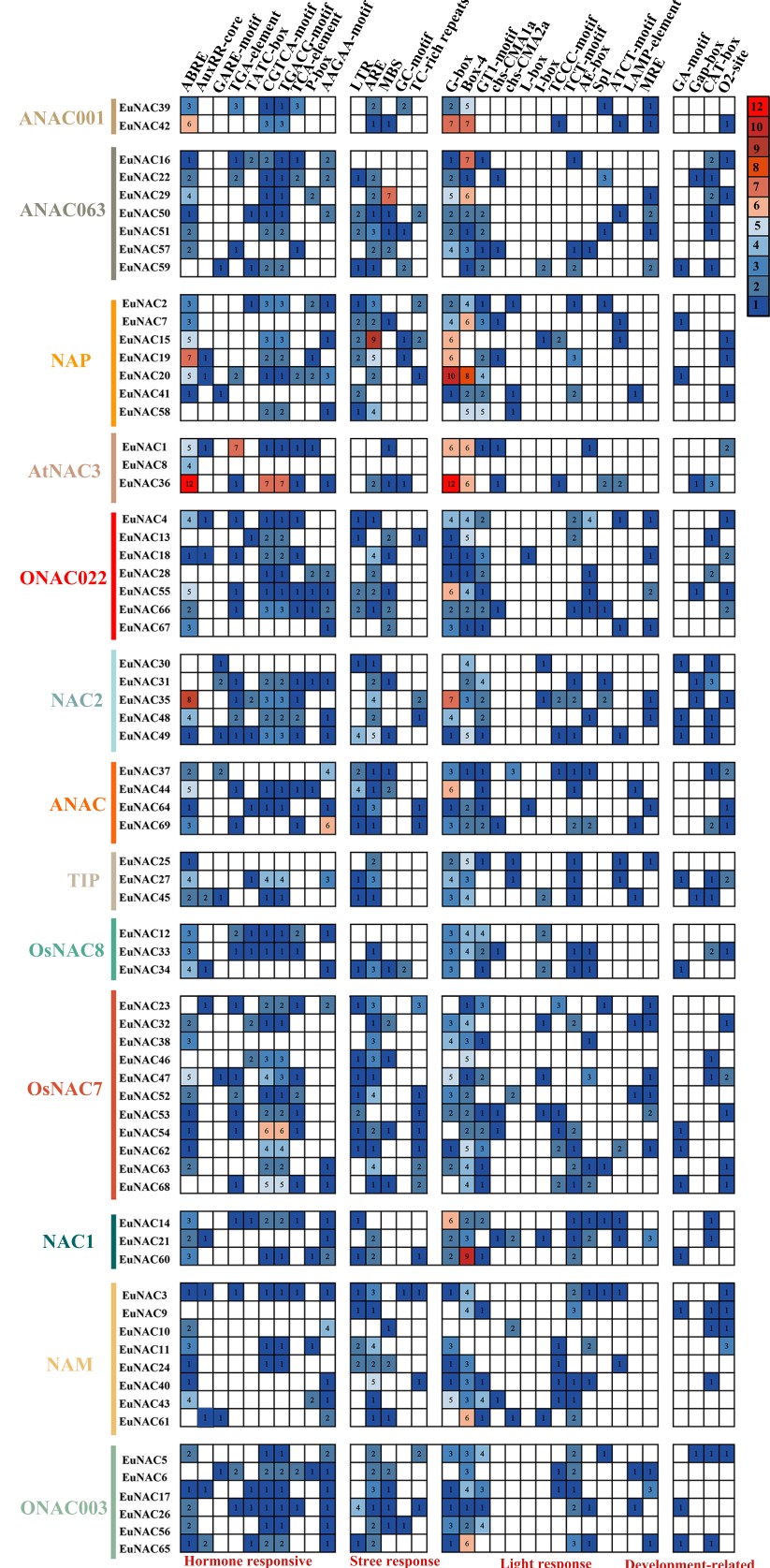

**Figure 6** The number of each type of cis-acting element in the promoter region of each *EuNAC* gene.

after drought stress treatment, and *EuNAC1*, *36*, *37*, *64*, and *66* were downregulated. The expression levels of *EuNAC8* and *13* first decreased and then increased with prolonged treatment time; in contrast, the expression level of *EuNAC61* first increased and then decreased (Fig. 7, Table S8). The variable expression patterns of *EuNACs* may indicate their differential roles in the drought stress response of *E. ulmoides*. In particular, differentially expressed *EuNACs* may play a crucial role in *E. ulmoides*' response to drought stress.

## DISCUSSION

The NAC transcription factor family is one of the largest gene families in plants. These factors are involved in regulating hormone signalling pathways, biotic and abiotic stress responses, and plant growth and development (*Xia et al., 2023*; *Yuan et al., 2020*; *Zhang et al., 2019*). Several plant genomes have defined the *NAC* gene family. The evolutionary connection and duplication patterns of the *NAC* gene family in *E. ulmoides* can be better understood thanks to the genome of this organism. Previous studies have shown that genes with close evolutionary relationships often share similar functions (*Lynch & Conery, 2000*). Therefore, by studying the evolutionary relationships across gene families, we can learn more about and perhaps even anticipate how genes operate (*Balazadeh et al., 2011*; *Zhang et al., 2019*).

According to our study, the genome of *E. ulmoides* included 69 *NAC* genes, which is fewer than that of *A. thaliana* (117 *NAC* genes) (*Ooka et al., 2003*), but similar to *K. obovate* (79 *NAC* genes) (*Du et al., 2022*). Our results indicate that the majority of EuNACs did not experience environmental selection-induced elimination, but rather demonstrated a high level of conservation throughout evolution, underlining the necessity for more research from an evolutionary standpoint. All 69 *NAC* proteins were divided into 13 subgroups based on their sequence homology and classification relative to *A. thaliana* (*Ooka et al., 2003*). *NACs* in *A. thaliana* exhibit a high degree of similarity among members of the same class or *NAC* subgroup. Four *EuNACs* in the *ANAC011* subgroup are orthologous to *A. thaliana* genes, including *AtNAC071* and *AtNAC096*, which are in charge of tissue reunification, dehydration, and other processes.

Our research indicates that the *NAM* subgroup has 8 *EuNACs* that are orthologous to *AtNAC054* and *AtNAC059* in *A. thaliana*, which are known to be crucial for organ development, programmed cell death, secondary wall building, and biotic and abiotic stress responses (*Kim, Kim & Park, 2007*). The seven *EuNACs* in subgroup *NAP* are orthologous to *AtNAC018*, *AtNAC025*, and *AtNAC56* and are essential for leaf senescence (*Guo & Gan, 2006*). Three genes of the *EuNAC* gene family's subgroup *TIP* are orthologs of the *A. thaliana* genes *AtNAC060* and *AtNAC091*. These orthologous genes have been demonstrated to be crucial in the stress response and abscisic acid (ABA) signalling (*Donze et al., 2014*; *Jeong et al., 2008*; *Li et al., 2014*). Four *EuNACs* in the *ANAC011* subgroup are orthologous to *A. thaliana* genes, including *AtNAC071* and *AtNAC096*, which are in charge of tissue reunification, dehydration, and osmotic stress (*Asahina et al., 2011*; *Yang et al., 2020*). Five orthologous *EuNACs* to *AtNAC016*, which are known to be involved in chlorophyll degradation, are found in the *NAC2* subgroup. This indicates that this

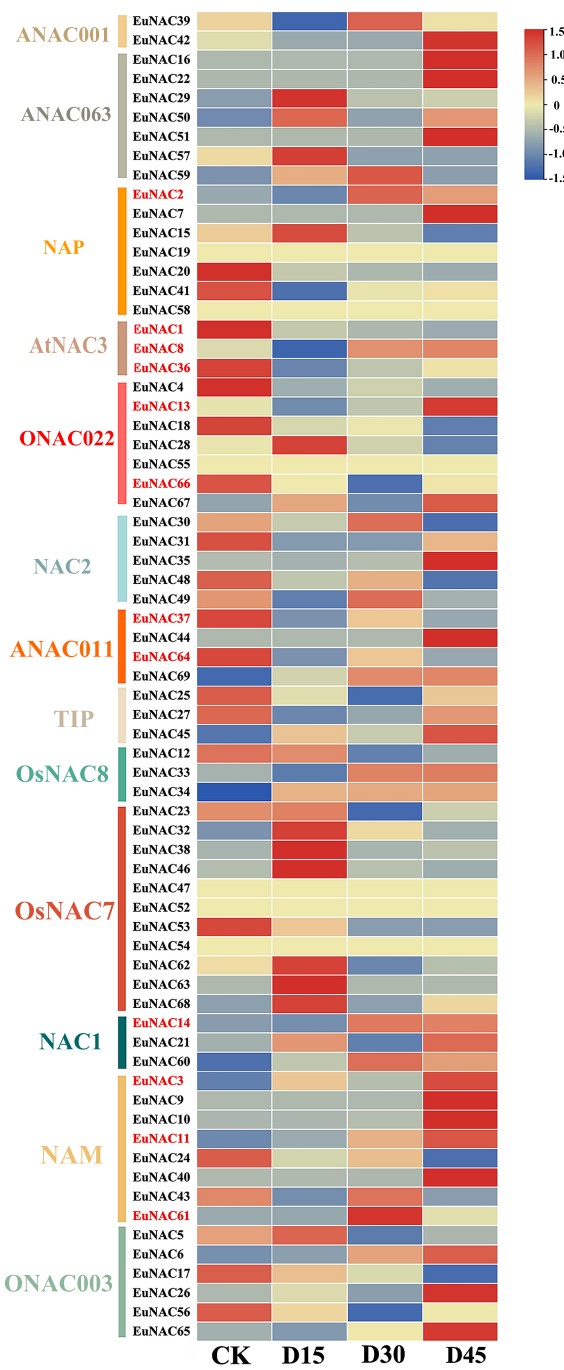

**Figure 7 Expression levels of 69 *NAC* genes under drought stress in *E. ulmoides* of leaves.** The expression level was presented based on the transformed data of log2 (FPKM+1) values.

subgroup of *EuNACs* may also control how chlorophyll degrades in plants (*Sakuraba et al., 2015*). Similar to *AtNAC003* and *AtNAC068*, the *ANAC001* subgroup also has two *EuNACs* that are orthologous to them. These genes control salt and osmotic stress tolerance, in addition to DNA damage responses (*Xu et al., 2013*; *Yoshiyama et al., 2014*).

Except for individuals from the *ONAC003* and *ANAC063* subgroups in this investigation, the N-terminus of the EuNAC protein had motifs 1, 2, 3, 4, 5, and 6. *ONAC7* comprised motifs 7 and 10, indicating that NAC transcription factors had a highly conserved N-terminus and a very varied C-terminus. The range of *EuNAC* introns was 2 to 8, which is comparable to the number observed in many plants (*Du et al., 2022*; *Liu et al., 2019*). In general, the deletion or insertion of introns can have diverse effects on gene function. Sometimes, intron deletions can lead to the loss of gene function if they result in a frameshift mutation or the removal of critical regulatory sequences. Our analysis supported the findings of Jeffares' research, which demonstrated that genes susceptible to abrupt changes in stress expression levels have much less intron density (*Jeffares, Penkett & Bähler, 2008*); *EuNACs* with fewer introns merit higher consideration if the study objective is to concentrate on genes that react instantly to environmental stress.

As a consequence of our findings, which included the identification of 12 segmental duplications and four tandem duplications in 69 *EuNACs*, we concluded that segmental duplication served as the primary catalyst for the growth of the *EuNAC* gene family, in agreement with research on *K. obovata* (*Du et al., 2022*). In addition, larger segmental duplications account for most of the *A. thaliana* genome, and at least four large-scale replication events occurred during the formation of angiosperm diversity (100–200 million years ago) (*Vision, Brown & Tanksley, 2000*). This might explain why there are more *NAC* members in *A. thaliana*, while having a smaller genome than *E. ulmoides*. Only 10 genes, according to our research, have collinear connections between *O. sativa* and *E. ulmoides*. However, we discovered 27 orthologous pairings in *A. thaliana*, a dicotyledonous plant. These findings demonstrate a closer evolutionary link between dicotyledons and *EuNACs* than between monocotyledons.

Cis-acting elements are specific DNA sequences located in the promoter region of genes that serve as binding sites for transcription factors (*Kaur et al., 2017*; *Liu, Sun & Wu, 2016*). In this study, more than half of the 69 *EuNAC* promoters included ABRE homeopathic elements, suggesting that these genes may operate *via* the control of ABA. MBS elements were found in 29 *EuNAC* s, indicating that these genes may be crucial in response to drought stress.

Research on gene function can benefit from understanding the patterns of gene expression. According to RNA-seq studies, drought stress drastically altered the expression levels of a few *EuNAC* s in the leaves of *E. ulmoides*. Significant differential expression was seen in 12 *EuNACs*. *EuNAC1*, *EuNAC8*, and *EuNAC36* are identical to *ANAC019* (*AT1G52890*), *ANAC055* (*AT3G15500*), and *ANAC072* (*AT4G27410*), which are members of the *AtNAC3* subgroup. Their expression is variably expressed during drought treatment, caused by drought, and stimulated by ABA (*Tran et al., 2004*). Therefore, we hypothesise that genes *EuNAC1*, *EuNAC8*, and *EuNAC36* belong to the same subgroup and are drought-responsive genes that control *E. ulmoides'* survival ability in drought-stressed

environments. Additionally, *ANAC054* (*At3g15170*) and *ANAC059* (*At3g29035*) were identical to *EuNAC3*, *EuNAC11*, and *EuNAC61* and were grouped into the *NAM* subgroup, suggesting that they may be crucial in *E. ulmoides*' response to drought stress.

## CONCLUSIONS

In summary, 69 *NACs* were identified in *E. ulmoides* in this study. We studied the characteristics of *EuNAC* genes at the genomic level and analysed expression patterns and responses to drought stress. These TFs can be divided into 13 subgroups according to the *NAC* classification method of *A. thaliana*. Chromosomal localisation and homology analysis showed that segmental duplication was the main driving force for *EuNAC* gene amplification. Our study provides the systematic information, functional framework, and expression patterns under drought stress about *EuNAC* genes, which will be facilitate the further functional studies of *EuNACs* in response to drought stress for *E. ulmoides*.

## ACKNOWLEDGEMENTS

The authors would like to thank YongJie Yang, GuoTao Song and YiXue Sun for their help in our experiment.

### Funding

This work was supported by the High level Talents Scientific Research Startup Project of Shihezi University (No. RCZK201948), the Talent Development Project of Xinjiang Production and Construction Corps (No. CZ004018/0205), the New variety cultivation project of Shihezi University (No. KX03090313), and the Self-Funded Project of Shihezi University (No. KX03100202). The funders had no role in study design, data collection and analysis, decision to publish, or preparation of the manuscript.

### Grant Disclosures

The following grant information was disclosed by the authors:
High level Talents Scientific Research Startup Project of Shihezi University: RCZK201948.
Talent Development Project of Xinjiang Production and Construction Corps: CZ004018/0205.
New variety cultivation project of Shihezi University: KX03090313.
Self-Funded Project of Shihezi University: KX03100202.

### Competing Interests

The authors declare there are no competing interests.

### Author Contributions

- Qi Wang conceived and designed the experiments, performed the experiments, analyzed the data, prepared figures and/or tables, and approved the final draft.

- FengCheng Hu conceived and designed the experiments, authored or reviewed drafts of the article, and approved the final draft.
- ZhaoQun Yao conceived and designed the experiments, performed the experiments, analyzed the data, authored or reviewed drafts of the article, and approved the final draft.
- XinFeng Zhao conceived and designed the experiments, authored or reviewed drafts of the article, and approved the final draft.
- GuangMing Chu conceived and designed the experiments, performed the experiments, prepared figures and/or tables, and approved the final draft.
- Jing Ye conceived and designed the experiments, performed the experiments, analyzed the data, prepared figures and/or tables, authored or reviewed drafts of the article, and approved the final draft.

## DNA Deposition

The following information was supplied regarding the deposition of DNA sequences:

The sequences of *Eucommia ulmoides*, *Arabidopsis*, and *Oryza sativa* used in this article are available at GenBank: GCA_016647705.1, GCA_000001735.2, and GCA_001433935.1.

## Data Availability

The raw measurements are available in the Supplementary Files.

## Supplemental Information

Supplemental information for this article can be found online at http://dx.doi.org/10.7717/peerj.16298#supplemental-information.

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
