# Peer review of "Comprehensive genomic characterisation of the NAC transcription factor family and its response to drought stress in Eucommia ulmoides"

_PeerJ, doi:10.7717/peerj.16298_

## Round 0.1 · original submission · Minor Revisions

Reviewers highlighted key points related to the manuscript's improvement. Is it possible to display vertically to make Figure 7 more readable? This is my suggestion to make it better. Waiting for your corrections.

Sincerely, good work

·

Basic reporting

The manuscript is concise, understandable and unambiguous.

Experimental design

no comment

Validity of the findings

no comment

Additional comments

I suggest making the following changes;
Line 84, Arabidopsis thaliana
Line 143; not TO, to
Line 154; not Arabidopsis, change it as A. thaliana
Line 155 add NAC classification system
Line 200; write A. thaliana and Oryza sativa
Line 202-203; A. thaliana and Oryza sativa
Line 204-205; Is E. ulmoides a dicotyledonous plant? it should be mentioned here.
Line 315-316; Is there any reference paper that these genes (ANAC054 (At3g15170) and ANAC059 316 (At3g29035) are involved in drought? Please give a reference here.
Line 321; Delete ‘tissue expression pattern’ because tissue expression patterns have not been determined in this study.

Reviewer 2 ·

Basic reporting

What is the aim of this study?
The article has been well-structured with a very coherent flow of information. Although the plant in focus is not extensively cultivated, the study's uniqueness and its contribution to science are evident, as it introduces a new plant to serve people or unveils the characteristics of this plant for the benefit of humanity. As a result, it constitutes an original and highly impactful scientific endeavor.

Experimental design

Line 101. İs it possible to do this analysis in MEGA 6 (MEGA 7, MEGA X) based on this sentence “neighbor-joining (NJ) method with 1000 bootstrap iterations”
Line 102. How did you do this classification, did you use the software?
As far as ı understand, Beijing genomic institute is out of order for genome-wide analysis. Last year ı called them and they gave this information. Pay attention to lines 135-136.
Line 134. Are you using liquid nitrogen for freezing or for physical grinding?

Validity of the findings

The references should be revised again to conform to the journal's guidelines. If possible, it would be beneficial to replace sources older than 23 years with more updated ones. Science is in a constant state of change and development, and consideration should be given to the possibility that older information might have lost its relevance. Proper citation of the Mega 6 program is necessary wherever referenced. Instead of providing extensive explanations under Fig.1, Fig.2, Fig.3, etc., it would be advisable to offer concise explanatory information, with the current figure details provided in the Materials and Methods section.
Genome-wide expression analysis of NAC genes in response to drought provides an opportunity to further understand the strong tolerance mechanism of E. ulmoides to drought.
For such an exceptional study, the conclusion and recommendation statements seem rather simplistic. In other words, using such expressions might not be warranted for a study of this nature. It would be more beneficial to phrase it in a more specific manner, highlighting a significant deduction drawn from the study.

Additional comments

Discussion section is well-structured.

Annotated reviews are not available for download in order to protect the identity of reviewers who chose to remain anonymous.

·

Basic reporting

In general, the use of English is clear, but it needs to be checked for some spelling errors, e.g., in line 210 Figure is misspelled, as in line 323 the word localization.
In the introduction, the use and importance of E. ulmoides and the importance of resistant cultivars to overcome numerous stresses are well defined. The involvement of NAC genes in the response to biotic and abiotic stresses in other plants is well established. The literature and references are sufficient to provide an overview of the purpose and hypotheses of the article.
The structure of the article is clear.
The quality of the figures in the article needs to be improved because they look blurry and become distorted when enlarged. Figure 7 is too small and of poor quality, making it difficult to understand the names of the genes.
I could not find the cDNA library that the authors submitted to the NCBI Sequence Read Archive.

Experimental design

Overall, the experimental design is well defined and logical. My only question is why the expression of NAC genes was done only in leaves and not in other tissues of the plant. Maybe you can explain this in the results.

Validity of the findings

The results of this research may lead to improvement of E. ulmoides varieties to overcome some stress factors. This article is a first step toward developing improved plants, although further research is needed to understand the importance and function of each NAC gene.

---

## Round 0.2 · accepted · Accept

Your manuscript has been accepted because you have improved it in accordance with the reviewers' and editor's comments. Congratulations

·

Basic reporting

The authors complied with the comments made.

Experimental design

The authors complied with the comments made.

Validity of the findings

The authors complied with the comments made.

Additional comments

The authors answered the doubts satisfactorily.